# Morphological Intervention in Promoting Higher-Order Reading Abilities among College-Level Second Language Learners

**Haomin Zhang [1,2,*]** and **Weicheng Zou [2]**

1   Faculty of Education, East China Normal University, Shanghai 200062, China
2   Department of English, School of Foreign Languages, East China Normal University, Shanghai 200241, China; wczou@english.ecnu.edu.cn
*   Correspondence: hmzhang@english.ecnu.edu.cn; Tel.: +86-21-54344893

**Abstract:** Reading success in a second language (L2) is vital to sustainable language and academic development because reading serves as a tool to absorb and learn new knowledge. Particularly in the context of college English as a foreign language (EFL), students constantly face the challenge to read English material to develop content knowledge. The current study investigated the effect of explicit morphological instruction on L2 students' higher-order inferencing and comprehension abilities. Sixty-two Chinese collegiate EFL students who were taking an intensive reading course (31 in the treatment class and 31 in the control class) participated in this study. The morphological intervention in the treatment class focused on identifying, decomposing, analyzing, associating, applying word parts in context. The control class received no explicit instruction in morphological awareness. After one semester of instruction, a series of morphology, inferencing and comprehension measures were administered to the participating students. The results showed that the didactic intervention of morphological awareness contributed to morphological knowledge and word-meaning inferencing ability, whereas there was no significant relationship between morphological intervention and text-based inference and comprehension abilities. The findings suggest that the intervention has a direct impact on word learning ability; however, higher-order processing skills may not directly benefit from it in a short period of time. Given that reading comprehension requires fine-tuned understandings of both local meanings and global contextual information, morphological awareness may not have an immediate effect on comprehension. Applied implications are also discussed in relation to effective morphological instruction and reading development in L2 contexts.

**Keywords:** morphology; Chinese EFL students; experimental and control; word learning; reading comprehension

## 1. Introduction

Sustainable development, to a great extent, highlights broad measurable elements (e.g., economic components); however, the core of sustainability is the advancement of human choices and human needs [1,2]. Through the inspection of those human-related or human-centered aspects of sustainability, it is found that enhanced human learning and communication are key components of sustainable development [2]. Language and literacy development play critical roles in fostering human-centered sustainability. Scholars underscore the role of sustainable reading development in empowering individuals and, at the same time, enhancing societal development [1–3]. In the current era of information, reading and literacy development are not restricted to the mother tongue, and development efforts are also devoted to facilitating biliteracy and second language (L2) reading development.

Reading acquisition is a very complex process because it necessitates fundamental semantic meaning activation at both the local word level and the global text level [4]. Reading researchers have shown that word learning and reading development benefit from a wide variety of cognitive and linguistic abilities (e.g., working memory, metacognitive strategies, word reading, lexical inference) [5,6]. The existing literature has extensively focused on the correlation between various indicators and reading outcomes, which lack an in-depth understanding of how various factors can contribute to actual instruction and learning. L2 learners struggle with word learning and reading success, given that a large number of lexical items are needed to comprehend authentic print material. Though we argue that word learning can undergo either implicit or explicit learning modules (intentional learning or incidental learning), words cannot be automatically stored in the learner's mind without explicit abstraction of meaning and use in context. L2 English teachers encourage students to be immersed in different resources to develop additional vocabulary knowledge outside of the classroom. The question remains as to how teachers can help students develop a vocabulary in class settings and incentivize students to use a specific skill to expand word knowledge on their own, thus contributing to their higher-level reading development.

Partial word knowledge has been found to help students break down words to extract key semantic information and retrieve meaning from printed words systematically [7]. Morpheme is a crucial component of partial word meaning that generates core semantic information within a large chunk of words. Morphology is an umbrella term for the study of words, word formation and structural regularities within words. In general, morphology can be divided into inflectional morphology (e.g., third- person singular- s), derivational morphology (e.g., prefixes and suffixes: teach-er, respect-able, il-legal, un-usual) and compounding morphology (e.g., fire-fighter). Chinese morphology highlights lexical compounding as the most dominant means of word formation [8]. Students with first language (L1) Chinese backgrounds have exposure to numerous lexical compounds, given that the majority of Chinese morphologically complex words are compound words (c.f. a lexical analysis in [9]). Morphology in Chinese forms the foundation for vocabulary acquisition among L1 children [6,10–12]. Before Chinese students start to learn L2 English, they are equipped with L1 morphological sensitivity and morphological awareness to further exploit L2 linguistic resources. Morphological awareness is the ability to manipulate and reflect upon morphological structures within complex words [13] and the explicit extraction ability is believed to predict reading abilities. A number of correlation-based studies have found that morphological awareness contributes to both L1 reading [6,14] and L2 reading [5,15,16] among learners with diverse linguistic and orthographic backgrounds. More strikingly, the extant studies have endorsed the cross-linguistic sharing of morphological awareness between two typologically and orthographically distant languages, i.e., Chinese and English [5,17]. However, the literature has emphasized monolingual and bilingual reading development among school-age children and reading abilities mostly focused on lower-level decoding and word recognition skills. To date, empirical studies lack a comprehensive investigation of the possible causality between morphological awareness and higher-order L2 reading abilities among college students. The current study aimed to explore whether explicit morphological awareness intervention in college-level L2 classrooms can help students develop higher-order word learning and reading abilities.

## 2. Review of Literature

As mentioned above, extant studies have systematically found that morphological awareness is related to diverse reading skills in different languages and orthographies. Additionally, a few studies made attempts to explore the instructional effect of morphological awareness on vocabulary and reading development through quasi-experimental analyses [18–20].

Baumann et al. conducted an instructional study investigating the effect of morphological and contextual analysis instruction on word-meaning inference ability among elementary-age students [20]. The instruction emphasized learning prefixes and suffixes (form and meaning) and more importantly,

the contextual use of target morphemic structures. The findings demonstrated that the treatment group outperformed the traditional vocabulary instruction group on inferring new affixed words and novel morphologically complex words. With regard to the reading measures, there were no differences between the treatment group and the control group on reading comprehension and content learning ability.

Bowers and Kirby implemented a 20-session morphological intervention focusing on words' morphological structures and aimed to explore the effect of morphological instruction on vocabulary knowledge among Grade 4 and 5 students [18]. They randomly assigned students to treatment and control groups and the instruction in the treatment group focused on the integration of spelling and morphology. To be more specific, students were explicitly taught to identify meaning cues in consistent spelling patterns. The findings showed that the students in the treatment class were more effective in using existing vocabulary information to develop new vocabulary knowledge.

More recently, Goodwin probed into the effectiveness of morphological awareness instruction in reading development in urban school students, including both L1 English students and language minority students [19]. The overall instructional goal focused on comprehension strategy. The control group concentrated solely on reading comprehension strategy teaching and learning, whereas the intervention group integrated both morphological problem-solving and comprehension strategy instruction. All the participating students had four 30-min small-group guided reading instructions with morphological instruction incorporated in the intervention group. The results showed that morphological instruction significantly and effectively enhanced vocabulary knowledge development. Nonetheless, no significant differences were found in reading comprehension and word reading fluency between the intervention group and the control group. More strikingly, the results indicated that the morphological instruction was particularly conducive to the literacy development of language minority students in the urban context.

The aforementioned studies have shown that morphological awareness instruction can significantly contribute to word learning and potential reading skills. Carlisle states that morphological awareness instruction will possibly contribute to reading development pertaining to three perspectives—sound, form and meaning [21]. However, it is unclear how morphological instruction should be implemented in L2 classrooms to support vocabulary and reading skills to a greater extent. Several aspects of good morphological awareness teaching are discussed [7]. First, teaching should take place in the context of rich and explicit vocabulary instruction. Morpheme knowledge acquisition should be situated in an environment of explicit vocabulary learning. It is also important to ensure that morphemes are introduced in meaningful contexts that students can comprehend. Second, teaching should help students use morphology as an explicit cognitive strategy to learn words. Morphemic forms and meanings are not only sets of rules to memorize words but systematic ways to develop word-meaning knowledge. Students are supposed to recognize and analyze word parts, infer and confirm word meanings based on the context. Third, teaching should relate to explicit morphological knowledge and its application in context. Along with the first two steps, morphological awareness instruction should integrate knowledge acquisition (knowing the meanings of prefixes, suffixes, roots) and adaptation (knowing how words are transformed: nominalization). Above all, morphological awareness instruction should be explicitly carried out in the classroom setting, helping students to learn word parts, recognize them, and apply them in context.

Based on the review and the discussion, a number of research gaps need further exploration. First, previous studies found that morphological awareness instruction can facilitate word knowledge and word learning ability, but higher-level reading ability is not necessarily influenced by morphological instruction. Second, most studies highlighted the effectiveness of morphological awareness instruction among L1 school-age students but overlooked the importance of morphological teaching in the L2 context. Thirdly, there were various ways of morphological awareness instruction and morphology learning. The design and the quality of instruction in morphology may generate different patterns of

development [21]. To address the existing empirical and practical questions, the current study aimed to explore one central question by a quasi-experimental research design:

To what extent can systematic morphological intervention help adult EFL students develop higher-order inferencing and comprehension abilities across time?

## 3. Method

### 3.1. Participants

Sixty-two first-year English major students participated in this study, with 31 in the treatment group (3 males and 28 females) and 31 (4 males and 27 females) in the control group. The participants were selected from four parallel first-semester intensive reading classes at a leading university in Shanghai, China. The participants had similar levels of prior academic achievement because they had either been pre-selected from top foreign language immersion schools or had passed through the college entrance examinations from top high schools throughout China. Before the semester started, the treatment class and the control class were randomly selected. One of the authors was the instructor of the treatment class and another experienced professor with 15 years university teaching experience taught the control class. Two instructors met on a regular basis to discuss lesson design, class activities and assessments. There were three overarching learning objectives for the entire four intensive reading classes: (1) developing a wide range of contextualized vocabulary knowledge in reading sources; (2) connecting source text information, background knowledge, emerging linguistic knowledge to achieve higher-level comprehension, and; (3) using source text information to expand, analyze and reflect upon familiar topics.

The study was exempt by the Institutional Review Board of the researchers' institution for the reason that the study involved adult participants in classroom settings and the assessments distributed to the participants were all used for the purpose of improving teaching and learning. The informed consent was obtained prior to data collection.

### 3.2. Intervention

As noted earlier, both treatment and control groups were from the first-semester intensive reading classes. The course was 18 weeks in length, covering seven thematic units in the textbook *Integrated Skills of English* [22]. Each unit was divided into three major phases: communicative activities, reading activities and extended activities. The communicative activities included discussion of topic-related questions prior to learning the new content. Main instruction and learning activities took place in the next two phases. The reading activities consisted of one main learning text. Text genres varied across different units (e.g., narrative episodes and informational texts). The extended activities included one supplementary reading material centering on the theme of the unit. In addition, oral and written assessments were conducted in the last phase. There were two 90-min class meetings each week, and each unit was completed within four class meetings.

For the treatment group, explicit morphological instruction modules were added in the reading and extended phases of each unit. The instruction modules focused on the words in the main text and the supplementary text. We adopted some key components of the Lesaux et al. intervention program [23] and refined our instructional approaches for the college-level EFL students. We would like to take the following segment from the text entitled "Genius Sacrificed for Failure" as an example. Table 1 presents the sequence of morphological instruction in this specific segment. First, students were instructed to identify and recognize new words with morphologically complex structures. Second, they were asked to break down the words into smaller segments based on their understanding. Third, the instructor confirmed the segmentation of morphological structures and taught the meanings of different word parts. Fourth, the students were asked to produce words with similar morphological structures based on their existing word knowledge. Finally, they were asked to apply the derived new words in sentential structure or context. The instructor also provided some additional examples to the students.

*Undaunted, they continued in their spare time, late at night by candlelight, to pour out their pent-up emotion, writing of what they knew best, of women in conflict with their natural desires and social condition- in reality, less fiction than autobiography!*

**Table 1.** Instructional sequence of the treatment group.

| Morphological instruction | |
|---|---|
| **Steps** | **Examples** |
| Identification of new words with morphologically complex structures. | Undaunted, autobiography |
| Segmentation of morphemic units in morphologically complex words. | *un*-daunt-*ed*, *auto-bio-graphy* |
| Learning morphemic meanings. | *un*-: not, opposite of, <br> *-ed*: past tense, past participle, participial adjective <br> *auto*-: self <br> *bio*-: life <br> *-graph*-: to write, written |
| Making associations based on similar word parts. | *un*-: unpleasant, unafraid, unfair, unconformable <br> *-ed*: surprised, tested, talked <br> *auto*-: autograph, automate, automobile <br> *bio*-: biology, biography, biome, biosphere <br> *-graph*-: photograph, geography, calligraphy, graphic, grapheme |
| Applying and using words in a sentence or context. | "My spirits were undaunted, and my commitment as firm as ever." <br> "Bill Clinton wrote about his experiences in his autobiography." <br> "Traditional tools are being replaced by automated machines." <br> "Fans are surging around the hotel to ask for autographs." |

*3.3. Data Collection Procedure*

After the one-semester instruction, five outcome measurements (described below) were administered to the participating students at the end of the semester. Different tasks were randomized and counterbalanced in the two classes. Data collection was conducted in a whole-class session. The total time allotment was 90 min.

*3.4. Measurement*

3.4.1. Pre-test Control Variables

**Vocabulary knowledge**

A pre-test vocabulary knowledge task was administered to the participants. The English Vocabulary Size Test [24] was used to monitor the students' initial language competence, thus creating the baseline for the later comparison. Sixty vocabulary items representing 6000 words were taken from six levels of vocabulary and the bilingual (English–Chinese) version was used. The reliability (Cronbach $\alpha$) was 0.812.

**Morphological knowledge**

The morpheme discrimination task adapted from Ku and Anderson [25] was used to measure the students' initial pre-test morphological knowledge. In each item, three possible derived or compound words were shown to the participants, all of which seemed to contain the same morpheme. The participants were asked to circle the word which did not fit the morphological pattern. For example, three words—estimate, classmate, and roommate—were presented to the participants. All the three words share the same subcomponent -mate, however the -mate in estimate does not carry the same morphemic meaning as the rest do. The participants were supposed to circle the outlier of the three words. The reliability (Cronbach $\alpha$) was 0.733.

3.4.2. Post-test Outcome Variables

**Morphological knowledge**

- *Morpheme form knowledge*

The morpheme form knowledge task was adopted from the Word Part Levels Test [26]. The participants were required to recognize word parts (both prefixes and suffixes) that change the meaning or the part of speech of a word. For example, four word parts with the same number of letters were presented to the participants: (1) sal-, (2) cau-, (3) lin-, and (4) dis-, and they were asked to choose the appropriate English affix. There were 40 items in this measurement. The reliability (Cronbach $\alpha$) was 0.703.

- *Morpheme meaning knowledge*

The morpheme meaning knowledge was also modeled after the Word Part Levels Test [26]. The participants were asked to choose the appropriate meanings based on presented affixes. For instance, an affix, *co-* (co-worker, co-exist), and four options, (1) person/thing, (2) direction, (3) together, (4) main, were shown to the participants and they needed to judge the appropriate meaning of *co-*. There were 34 items in the morpheme meaning knowledge measurement. The reliability (Cronbach $\alpha$) was 0.775.

**Inferencing and comprehension abilities**

- *Pseudoword inference*

The pseudoword inference task was adopted from Guessing from Context Test [26]. The task asked the participants to guess the meaning of an underlined word in context (sentences or short passages). For example, a sentence and three options were shown to the participants—"The fish used to be cheap, but it is very gloch now," with options: (1) fast, (2) rich, (3) expensive. The underlined word—*gloch*—was a pseudoword that required the participants to make an inference. Based on the contextual information and constraints, the participants were supposed to choose the most accurate meaning (*expensive*, in this example). There were 20 items in this measurement. The reliability (Cronbach $\alpha$) was 0.798.

- *Real word inference*

The cloze test measured the learner's ability to comprehend context to fill in missing lexical items. There were two texts in this measure and 10 real words were deleted in each text. The reading material was selected from a collegiate English major reading exercise book. The participants were asked to choose one appropriate lexical item from the word list below each text. The word list consisted of 10 correct responses and five distractors. In total, there were 20 items in the cloze test. The reliability (Cronbach $\alpha$) was 0.771.

- *Text comprehension*

The text comprehension required learners to identify specific information, make inferences, and detect main ideas. Similarly, the reading texts were chosen from the reading exercise book. There were four texts with five corresponding comprehension questions in each text. The participants needed to choose the most accurate response from four options given below each question. In total, there were 20 items in the text comprehension task. The reliability (Cronbach $\alpha$) was 0.813.

## 4. Results

*4.1. Descriptive analyses of the pre-test and post-test results*

The descriptive data in Table 2 show the average score of each measurement performed by the two groups in the pre-test and the post-test. In the pre-test control measurements, morphological

knowledge and vocabulary knowledge had adequate dispersion based on standard deviations. The mean differences of the two tasks did not reach significance level: morphological knowledge, $t(60) = 1.62$, $p = 0.976$; vocabulary knowledge, $t(60) = 0.031$, $p = 0.113$. The results indicated that initial language competence and morphological knowledge were similar, which created the basis for post-test comparisons.

**Table 2.** Descriptive statistics for student performance in vocabulary and reading in both treatment and control groups.

| Group | Measurement | Mean | Std. Deviation |
|---|---|---|---|
| | *Pre-test* | | |
| | Morphological knowledge (20) | 18.40 | 1.63 |
| | Vocabulary knowledge (60) | 41.52 | 5.40 |
| | *Post-test* | | |
| Treatment (*n* = 31) | Morpheme form (40) | 37.87 | 3.43 |
| | Morpheme meaning (34) | 32.68 | 1.30 |
| | Pseudoword inference (20) | 16.35 | 1.85 |
| | Real word inference (20) | 14.87 | 2.53 |
| | Text comprehension (20) | 14.84 | 1.79 |
| | *Pre-test* | | |
| | Morphological knowledge (20) | 18.39 | 1.76 |
| | Vocabulary knowledge (60) | 39.45 | 4.65 |
| | *Post-test* | | |
| Control (*n* = 31) | Morpheme form (40) | 35.00 | 6.35 |
| | Morpheme meaning (34) | 31.03 | 2.55 |
| | Pseudoword inference (20) | 15.10 | 2.93 |
| | Real word inference (20) | 14.48 | 2.53 |
| | Text comprehension (20) | 14.29 | 2.30 |

In the post-test outcome measurement, based on the standard deviations, we found that the scores of morpheme meaning and morpheme form varied more widely in the control group. In general, the treatment group performed better in the semester-end outcome measurements. Table 3 presents the results of correlational analysis of outcome measures in both the treatment and the control groups. The results demonstrated that morphological knowledge dimensions were all significantly correlated with inferencing and comprehension outcomes in the treatment class, with coefficients ranging from $r = 0.34$, $p < 0.05$ to $r = 0.78$, $p < 0.001$. However, morphological knowledge dimensions had weak or no significant relationships with inferencing and comprehension measurements. Both morphological knowledge dimensions were not significantly related to pseudoword inference, $r = 0.19$, $p = 0.313$; $r = 0.19$, $p = 0.318$, and they were mildly related to real word inference and text comprehension, with coefficients ranging from $r = 0.35$, $p < 0.05$ to $r = 0.54$, $p < 0.05$.

**Table 3.** Bivariate correlations among post-test outcome measures in the treatment group and the control group.

| Variable | 1 | 2 | 3 | 4 | 5 |
|---|---|---|---|---|---|
| *N = 31 (treatment group above the diagonal)/N = 31 (control group below the diagonal)* | | | | | |
| 1. Morpheme form | - | 0.44 * | 0.61 *** | 0.78 *** | 0.66 *** |
| 2. Morpheme meaning | 0.35 * | - | 0.45 * | 0.42 * | 0.34 * |
| 3. Pseudoword inference | 0.19 | 0.19 | - | 0.68 *** | 0.55 ** |
| 4. Real word inference | 0.54 ** | 0.44 * | 0.35 * | - | 0.59 ** |
| 5. Text comprehension | 0.52 ** | 0.44 * | 0.37 ** | 0.55 ** | - |

*\* p < 0.05; \*\* p < 0.01; \*\*\* p < 0.001*

### 4.2. MANOVA analyses of outcome variables

Figure 1 presents the accuracy score of each measurement between the two groups. In line with the descriptive analysis, the findings revealed that the treatment class scored higher in all the five semester-end post-test measurements. A MANOVA test was conducted to examine the effectiveness of morphological intervention in higher-order inferencing and comprehension skills by comparing the treatment group and the control group. The test for homogeneity of variance–covariance (Box's test) was initially employed to check the central assumption of the MANOVA test. The result showed that Box's test was significant at $p = 0.004$. Given that there was unequal variance between the two groups, a more robust MANOVA statistic, Pillai's trace, was used in the following data analysis. The multivariate result demonstrated that there was a statistically significant difference in outcome measurements (including morphological knowledge measurements, pseudoword inference, real word inference and text comprehension) based on the group: $F (1, 59) = 2.71$, $p = 0.029$; Pillai's trace $= 0.198$, $\eta_p^2 = 0.198$. The further between-subjects analysis found that the treatment group outperformed the control group on morphological knowledge measurements, morpheme form: $F (1, 59) = 4.87$, $p = 0.031$, $\eta_p^2 = 0.076$; morpheme meaning: $F (1, 59) = 9.27$, $p = 0.003$, $\eta_p^2 = 0.136$. Furthermore, the results indicated that there was a marginally significant difference between the two groups on pseudoword inference, $F (1, 59) = 3.70$, $p = 0.054$, $\eta_p^2 = 0.063$. Nonetheless, the findings did not verify the significant intervention effect on real word inference and text comprehension: real word inference, $F (1, 59) = 0.269$, $p = 0.606$, $\eta_p^2 = 0.005$; text comprehension, $F (1, 59) = 3.89$, $p = 0.344$, $\eta_p^2 = 0.015$.

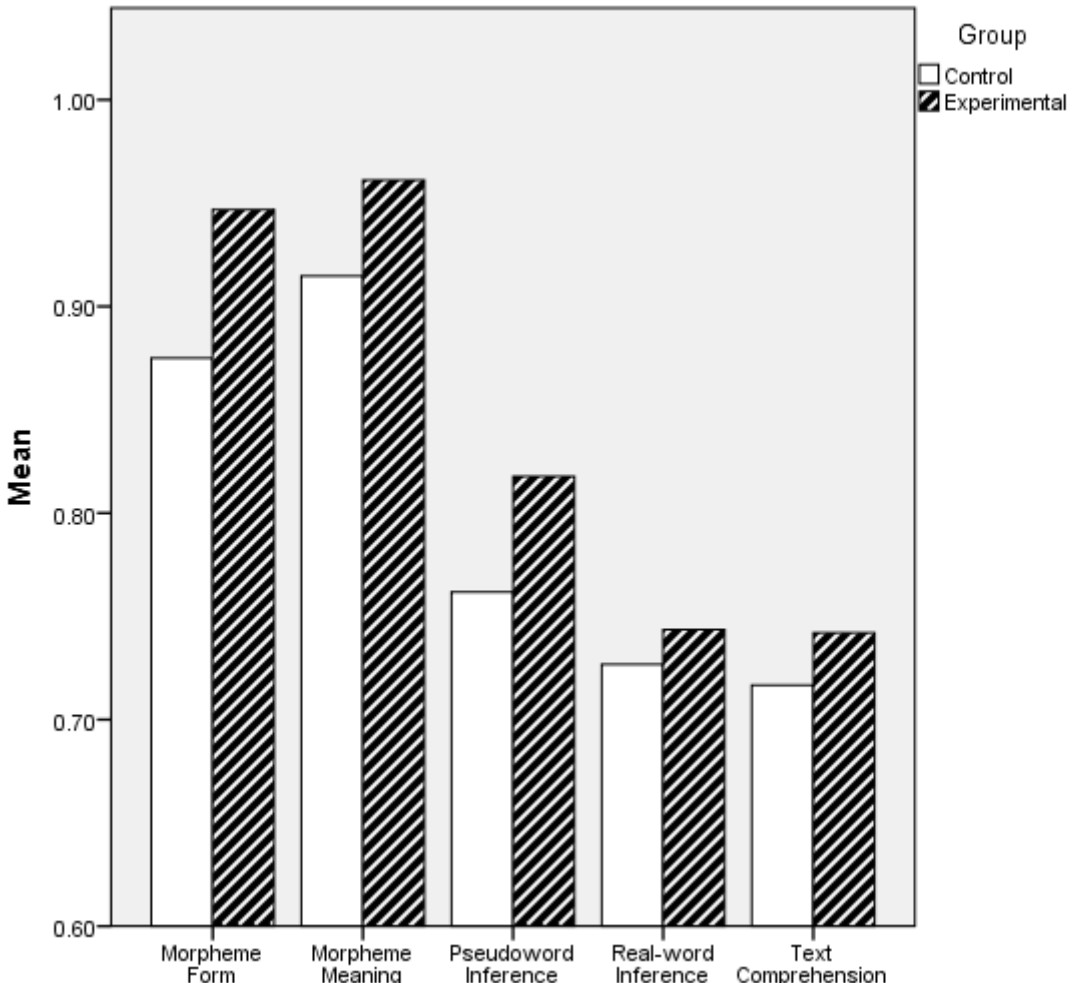

**Figure 1.** Accuracy rates of morphological knowledge, word-meaning inference and text comprehension.

## 5. Discussion

Under the quasi-experimental design, the current study yielded some findings that require further interpretation. First, the current study substantiated the existing findings of L1 studies that morphological awareness instruction could foster the development of word learning ability [18–20]. As evidenced in the positive and significant changes in morphological knowledge and pseudoword inference, the instruction to some extent helped college EFL students raise their awareness of partial word information, and more critically, their ability to develop word learning ability. The instruction focused on the role of morphemic knowledge, morphological awareness, and morphological applications in context, thus better shaping the students' sensitivity to segmental morphological information within complex words. From anecdotal experiences, Chinese EFL students are traditionally required to memorize a large number of words on their own. Lack of systematic assistance or instruction somehow increases the students' learning demands, and rote memorization would negatively affect the students' learning motivation. The instruction that the students received in the current study encouraged them to break down lexical bundles and make systematic associations by themselves. One of the benefits of teaching morphology is to motivate students to partake in word learning activities spontaneously through morphological problem solving [18]. Through systematic instruction, the participating students were gradually equipped with the ability to learn novel words in context.

Furthermore, it is crucial to interpret the patterns of higher-order processing skills between the treatment group and the control group. As discussed above, morphological instruction that helps to fill semantic gaps in context should support vocabulary development and potentially enhance reading comprehension [27]. The comparison analysis of the current study demonstrated that the treatment group had some advantages in text-based inference and comprehension but the differences were not salient. The first possibility may derive from the mechanism of reading comprehension development. Kintsch's construction and integration model [28] proposes that reading comprehension depends on three phases of knowledge building: word meanings, propositions and elaborations. Morphological assistance is conducive to the construction of local word meanings; nevertheless, propositions and elaborations that build upon textual information and background knowledge may not directly benefit from it [19,20]. The modality of the pseudoword inference task assessed the students' ability to construct local word meanings by inferring meanings at the sentence level. Nevertheless, real word inference and text comprehension abilities were measured in longer textual situations. In addition to local word-meaning construction, the real-word inference and text comprehension measurements necessitate an integrated understanding of syntactic and semantic networks as well as contextual sensitivity. Therefore, it is plausible that morphological intervention may not make a substantial contribution to higher-order text-based inference and comprehension across time. The second possibility would be the students' responsiveness to the morphological intervention. The findings demonstrated that students in the treatment group were generally responsive to the instruction, given their significant improvements in measured morphological knowledge. However, the outcome measurements capitalized on explicit but decontextualized morphological knowledge and did not fully capture their analytical ability in context. Morphological awareness/knowledge by itself involves the explicit abstraction of partial word form and meaning [29], which occurs mostly in a decontextualized fashion, whereas morphological analysis taps into the learner's ability to use specific word parts to retrieve the whole-word meaning [30]. Although the intervention covered both morphological awareness building and morphological analysis in context, the participating students may have been responsive to the straightforward decomposition and identification of morphemic cues rather than to the complicated morphological applications and use in context. Meanwhile, two different teachers in the treatment and the control groups could create an internal confound for the comparison of the improvements. Students may be more responsive to one specific instruction even without explicit morphological intervention. Additionally, the non-significance may also be affected by the cross-linguistic difference of L1 Chinese and L2 English. As mentioned above, cross-linguistic morphological awareness can be shared across languages to enhance reading acquisition in multiple languages. However, there are universal and language-specific

features underlying morphological processing [31]. Due to their L1 background, Chinese students may be aware of lexical compounding or boundaries in word formation, however, morphological intervention in the current study underlines the extraction and integration of derivational morphemes in English. Students may not recall their preexisting L1 experiences to further enhance L2 reading skills. Finally, the insignificant pattern may be attributed to the short time interval between instruction and outcome assessments. Although we did witness some positive changes from the outcome measures, reading success may not be achieved within a short period of time. Along with the discussion of the students' responsiveness to instruction, the semester-long instruction may not be sufficient to trace the ultimate success in reading comprehension, and students may need to adapt to the comprehensive process to conduct morphological analysis in context. Longer intervals would be ideal to scrutinize the actualization of morphological intervention and its connection with the developmental trajectory of higher-order inferencing and comprehension abilities.

## 6. Conclusions and Implications

The major objective of the current study was to substantiate the effectiveness of morphological intervention in L2 inferencing and comprehension abilities. The results showed a significantly positive effect on word-meaning inference ability but not necessarily on text-based inference and comprehension abilities. A few applied implications can be drawn from the findings. First of all, the morphological instruction of the current study was found to facilitate word-meaning inference ability (in the form of pseudoword inference). Systematic and explicit instruction in morphological awareness seems to support word learning ability [7]. One important implication for teaching practitioners is that teaching should be explicit and context-bound. It is crucial to explicitly teach students morphological knowledge (suffixes, prefixes and roots) and to raise the students' awareness of morphological structures in complex words. Meanwhile, it is extremely important to teach words or word parts in context so that students can relate to them. Furthermore, the current study did not attest to the significant relationship between morphological instruction and higher-order processing skills over a semester, although there were positive increases in text-based inference and comprehension. The major takeaway from this finding is the importance of long-term teaching and learning. The effectiveness of morphological instruction on higher-order reading development over the long term is not guaranteed, given that morphological awareness fills the semantic gap at the word level and higher-order reading abilities encompass a wide range of linguistic and cognitive abilities (local meaning extraction and global text-based meaning construction). Regarding instruction praxis, instructors would do well to encourage students to spontaneously learn more words and read more authentic materials over time. It is noteworthy that higher-order reading development would stem from long-term contextualized and meaningful instruction. The effects of morphological instruction on reading outcomes differ when morphology is taught in isolation or with other literacy skills and the effects can be sustained and transferred over a long period when morphological instruction is integrated with other aspects of literacy teaching [32]. Sustainable reading development in L2 is an intricate process because it involves cognitive and linguistic sophistication in various aspects. As utilities of morphological awareness are identified and verified, teachers ought to refine and proceed with the instruction for a longer period of time, thus enhancing the sustainability of higher-order reading acquisition.

**Author Contributions:** Conceptualization, H.Z. and W.Z.; methodology, H.Z.; validation, H.Z. and W.Z.; formal analysis, H.Z.; investigation, H.Z..; resources, W.Z.; data curation, H.Z.; writing—original draft preparation, H.Z.; writing—review and editing, H.Z.; supervision, H.Z.; project administration, H.Z.; funding acquisition, H.Z. All authors have read and agreed to the published version of the manuscript.

**Funding:** The study was funded by Shanghai Planning Project of Philosophy and Social Sciences (Grant No.:2018EYY009) and the Fundamental Research Funds for the Central Universities (Grant No.: 41300-20101-222397, 2018ECNU-QKT015).

**Conflicts of Interest:** The authors declare no conflict of interest. The funders had no role in the design of the study; in the collection, analyses, or interpretation of data; in the writing of the manuscript, and in the decision to publish the results.

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
