# Peer review of "Morphological Intervention in Promoting Higher-Order Reading Abilities among College-Level Second Language Learners"

_sustainability, doi:10.3390/su12041465_

Round 1

Reviewer 1 Report

The authors are to be commended for putting together a robust empirical study with practical implications regarding relationships between morphological instruction and improved language learning, implying that explicit applications of linguistic theory in the classroom can be beneficial for language learning. The study replicates L1 findings in an L2 context of a correlation between enhanced morphological awareness and enhanced lexical acquisition. This is a helpful contribution. The authors also argue that their results demonstrate the importance of contexualization for linguistic data when training L2 students to become more aware of morphological structures and processes. This too has important implications for EFL pedagogy.

My only concerns with the paper are relatively minor but important for the authors to consider in producing a lightly revised version of the article. These issues cluster around four concerns: (1) the need for discussion of human ethics review board approval and protocols, (2) the need for discussion of comparative morphological typology, (3) the need for more careful clarification of critical passages, and (4) the need for more careful general English language editing.

The first concern should be self-explanatory. This can be added to methods or mentioned in a footnote, depending on the preference of the journal. It is now standard practice to include such statements (often with registered clearance case number) for any study involving human subjects.

The second concern could be worked into the introduction at least and perhaps the discussion as well. Since English morphology and Mandarin morphology are quite different in terms of their respective typological profiles (Mandarin being more isolating/analytic, English being more agglutinative/concatenating), the importance of tuning in to English morphology becomes all the more acute for EFL learners whose mother tongue is Mandarin.

The third concern requires a careful re-reading of the text to ensure that key phrases and transitions are lucidly communicated. Did the authors investigate "the instructional effect of morphology on L2" (as stated in line 12) or did they investigate "effects of explicit training in morphological awareness"? Did the "results show that morphological intervention contributed to morphological knowledge..." (as stated in line 20) or did the results show that "didactic intervention focused on morphological awareness" contributed to morphological knowledge... ? In line 24, the "because" clause does not seem to be part of the findings, but rather an interpretation of the findings. A new sentence is needed to clarify this distinction. Given the importance of "morpheme" for the paper, a brief critical review of the literature on definitions of a morpheme, with a more clear, motivated definition proposed by the authors drawn from this literature, would be a welcome addition to the p.2 discussion (see lines 49-53). In line 153, was "an explicit morphological awareness instruction" added or were "explicit morphological instruction modules" added? In line 154, was the "instruction focused on the words" or were the "instruction modules focused on the words"? In line 179 is it the case that "A pre-test vocabulary knowledge was administered" or that "A pre-test vocabulary knowledge task was administered"? In line 349, is it "Through the instruction" or "In terms of instructional praxis"? In lines 349-350, the meaning of the following statement is simply unclear: "Morphological instruction, as an instructional tool or strategy, may not guarantee its effectiveness upon the implementation." Do the authors mean to say that "The effectiveness of morphological instruction over the long term is not guaranteed"? Please carefully and critically clarify this important closing point.

The final concern requires more careful proofing for English grammar in some places; though the paper is in good shape in this regard over all. Here is a non-exhaustive list of spots to check:

Reading success in second language >> Reading success in a second language students who was taking >> students who were taking After one-semester instruction >> After one-semester of instruction learning ability, however, higher-order >> learning ability; however, higher-order Morpheme is the crucial component >> The Morpheme is a crucial component Method >> Methods (see, e.g., line 126) Under the quasi-experimental design, >> Using a quasi-experimental design, findings that need further interpretations. >> findings that require further interpretation. Through the systematic instruction >> Through systematic instruction word meanings, nevertheless, propositions >> word meanings; nevertheless, propositions explicitly teach students’ morphological knowledge >> explicitly teach students morphological knowledge so that students can relate to. >> so that students can relate to them. Through the instruction, it is more advisable to >> Instructors would do well to

Incorporating these revisions should turn this into a fine and useful publication.

Author Response

Dear reviewer, thanks so much for raising four important comments. Changes to the ms were highlighted in red. Please feel free to let us know if additional clarifications would be needed. 

Discussion of human ethics review board has been added to the participants’ description. Thanks for raising the question about the typological difference. We have added explicit descriptions about the typological difference of morphology in Chinese and English to the introduction section. Meanwhile, we also added explicit discussion of cross-linguistic differences in reading development in the discussion section. We apologize for the confusion. We have edited all the questionable statements throughout the ms. To be more precise, we have clarified in the introduction section about the definitions of morpheme and morphology in two different languages (Chinese and English). In addition, we also edited the concluding paragraph. Thanks for checking the linguistic errors. We have edited through the entire manuscript and fixed all the errors.

Reviewer 2 Report

The paper is interesting, nicely presented but the section of CONCLUSIONS need to be extended,

There are many results that have been explained in the discussion section and need to be interelated to the literature to conform the conclusions. 

At the end of the article there are two extra pages that should not be there (pages 14-15).

Thank you

Author Response

Dear reviewer, thanks so much for raising the important comments. Changes to the ms were highlighted in red. We have added more in-depth interpretations in the discussion section and extended the conclusion section (highlighted in red).